# Compact Miniaturized Bioluminescence Sensor Based on Continuous Air-Segmented Flow for Real-Time Monitoring: Application to Bile Salt Hydrolase (BSH) Activity and ATP Detection in Biological Fluids

Aldo Roda [1,2], Pierpaolo Greco [3], Patrizia Simoni [4], Valentina Marassi [1], Giada Moroni [5], Antimo Gioiello [5] and Barbara Roda [1,2,*]

1    Department of Chemistry Giacomo Ciamician, Alma Mater Studiorum-University of Bologna, 40126 Bologna, Italy; aldo.roda@unibo.it (A.R.); valentina.marassi2@unibo.it (V.M.)
2    INBB—Biostructures and Biosystems National Institute, 00136 Rome, Italy
3    Department of Life Sciences, University of Modena and Reggio Emilia, 41121 Modena, Italy; pgreco8@gmail.com
4    Department of Medical and Surgical Sciences, Alma Mater Studiorum-University of Bologna, 40126 Bologna, Italy; patrizia.simoni@unibo.it
5    Department of Pharmaceutical Sciences, University of Perugia, 06123 Perugia, Italy; giada.moroni@chimfarm.unipg.it (G.M.); antimo.gioiello@unipg.it (A.G.)
*    Correspondence: barbara.roda@unibo.it

**Abstract:** A simple and versatile continuous air-segmented flow sensor using immobilized luciferase was designed as a general miniaturized platform based on sensitive biochemiluminescence detection. The device uses miniaturized microperistaltic pumps to deliver flows and compact sensitive light imaging detectors based on BI-CMOS (smartphone camera) or CCD technology. The low-cost components and power supply make it suitable as out-lab device at point of need to monitor kinetic-related processes or ex vivo dynamic events. A nylon6 flat spiral carrying immobilized luciferase was placed in front of the detector in lensless mode using a fiber optic tapered faceplate. ATP was measured in samples collected by microdialysis from rat brain with detecting levels as low as 0.4 fmoles. The same immobilized luciferase was also used for the evaluation of bile salt hydrolase (BSH) activity in intestinal microbiota. An aminoluciferin was conjugatated with chenodeoxycholic acid forming the amide derivative aLuc-CDCA. The hydrolysis of the aLuc-CDCA probe by BSH releases free uncaged aminoluciferin which is the active substrate for luciferase leading to light emission. This method can detect as low as 0.5 mM of aLuc-CDCA, so it can be used on real faecal human samples to study BSH activity and its modulation by diseases and drugs.

**Keywords:** bioluminescence; luciferase; aminoluciferin; ATP; bile acids; bile salt hydrolase-BSH; sensors; continuous flow assay

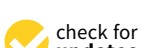



## 1. Introduction

The development of biosensors and devices for the continuous monitoring of analytes in a given fluid is an important analytical challenge. Obtaining sensitive real-time information is crucial to intervening promptly and effectively.

Many biological applications require the continuous in vivo and ex vivo monitoring of clinical biomarkers. These applications include quantifying glucose in diabetic patients, closed-loop drug delivery systems for anesthetic infusion, and observation of fast cellular events [1]. In addition, the continuous monitoring of analytes in a given fluid is important to achieve fast on-line control of industrial, environmental, and food science processes, such as tap water safety. These systems must offer high sensitivity and detectability to allow early diagnosis and prompt intervention. Specificity is also important in selecting the

appropriate actions at a relatively low cost, while retaining sufficient robustness. Moreover, ensuring sensitivity with robust protocols represents a challenging task when combined with requirements imposed by the market on low costs and sustainability for materials and operations.

Microfluidics, flow injection analysis (FIA), and compact lab-on-a-chip systems are widely used for these purposes [2]. These miniaturized systems offer several benefits over their larger counterparts. These benefits include smaller sample volume, reduced consumption of reagents, smaller size, potentially lower cost, and thus portability. These systems can also run multiplexed processes, increasing throughput and analytical performances.

The most popular devices for continuous monitoring are based on an FIA format, where the sample enters a microfluidic device and, once mixed with the appropriate reagents, is detected via colorimetry, luminescence, or electrochemistry [3].

FIA was introduced in 1975 by Ruzicka and Hansen [4], inspired by the autoanalyzer technique invented by Skeggs in the early 1950s [5]. In FIA systems, each analyzed sample is separated from the next by a carrier reagent. In contrast, the Skeggs' AutoAnalyzer used air segmentation to separate a flowing stream into numerous discrete segments, creating a train of individual samples moving through a flow channel. The automatic system commercialized by Technicon revolutionized the clinical chemistry laboratories for many years and it is now replaced by robotic discrete systems.

Similar approaches based on airflow systems were integrated with microfluidics-assisted technology and different formats to provide benchtop analyzers and, more recently, lab-on-a-chip analyzers.

These variants are more suitable than other portable simple biosensor formats for a near-real-time response. This is because they can continuously monitor a given process, rather than detecting the analyte with a high-productivity autoanalyzer.

Previously, we developed benchtop manifolds based on continuous air-segmented flow systems and various biochemiluminescence detection principles and formats [6,7]. Different analyte-specific enzymes (i.e., 3$\alpha$-hydroxysteroid dehydrogenase, lactate dehydrogenase, phenyalanine dehydrogenase) were immobilized on Nylon 6 tubes and coupled to bacterial luciferase enzymes to detect bile acids (BAs), lactate, and phenylalanine, respectively [8,9]. The measurements were based on the NADH cofactor, which is the substrate for the Lux bacterial luciferase: FMN-NADH oxidoreductase system [10]. Firefly luciferase was immobilized to detect ATP, ADP, and kinases while other enzymes (usually oxidases) were co-immobilized with peroxidase (HRP) or in a separate reactor to detect the produced $H_2O_2$ using luminol/enhancer as a chemiluminescence substrate [11,12].

These systems can perform up to 20 precise and robust analyses/hour at relatively low cost because the immobilized enzyme can be reused for up to 50–100 cycles [13].

In light of these successful applications, we used this principle to develop and design a new generation of a miniaturized continuous air-segmented flow devices and biosensors. In particular, the new generation of miniaturized light detectors (e.g., CCD, BI-CMOS) can image and quantify a very low amount of photons. Biochemiluminescence (BL and CL) reagents also have improved light efficiency emission.

BL and CL use a fast analytical light emission signal that can be measured without complex instrumentation. They can be directly recorded by collecting all the emitted light in all directions of the emitting area, without any chamber or geometrical limitations. Nowadays, the light detection or imaging has greatly improved relative to previous applications, which used a common photomultiplier tube (PMT) detector. Even though micro PMTs have been developed, light is mainly detected with CCD or CMOS-based devices. The back-illuminated CMOS of a smartphone or a conventional camera have demonstrated impressive performance and, thanks to their small size and low power demand, they are the best choice for miniaturized devices with a low voltage supply [14–17].

Device miniaturization technology has also improved greatly. Microperistaltic pumps and electrovalves can be used in compact portable devices for air-segmented continuous flow analyses.

There is growing interest in using in vivo or ex vivo biosensors to continuously measure the concentration of substances of pathological or therapeutic relevance in biological fluids such as blood, sweat, tears, microfiltrates, and cell cultures. Many sensors have been reported as potential implants [18–20]. Although they are a considerable scientific achievement, they also present a risk to the patient in terms of invasiveness and biocompatibility. In addition, problems of sensitivity and selectivity arise when the biosensor is implanted directly into biological fluids. These systems may improve the collection of biological fluids making the implantation into biological fluids simpler and more direct, resulting in high sample recovery and analytical throughput. In addition, the miniaturized systems allow for an in-situ analysis, i.e., at animal facilities, avoiding issues related to animal transportation such as sample contamination and time-consuming operations.

The combination of microdialysis sample collection and microfluidics is therefore important. Previously, sample size and sensitivity created limitations in coupling these systems. These limitations can now be resolved with highly miniaturized systems that require just a few microliters of sample and that are small enough to place on a finger.

Here, we report the development of a bioluminescence-based miniaturized sensing platform for continuous air-segmented flow assays. It is designed for use as a general platform for BL/CL-based detections, and is simple, versatile, modular, and robust. The prototype was built using miniaturized peristaltic pumps, low-diameter tubing, and specific enzyme-immobilized reactors. An integrated CMOS/CCD camera images the BL signal, which is processed with suitable imaging software.

Traveling in an air-water-air train, the sample is mixed with the appropriate reagents and reaches the enzymatic reactor, a Nylon 6 tube, where analyte-specific enzymes are immobilized. The tube is placed in front of the light detector, and a tapered face plate is used for contact lensless imaging and to accurately measure all the emitted light.

As a proof of concept to demonstrate the system's broad applicability, we developed two bioassay formats.

The first bioassay uses a microdialysis device and Nylon6 immobilized firefly luciferase to ex vivo measure ATP in the brain's extracellular environment including the caudate nucleus, jugular vein, and cerebrospinal fluid in the perfusate fluid.

The second approach uses the same BL system with Nylon 6-immobilized luciferase, but the substrate cocktail is modified to evaluate microbiome bile salt hydrolase (BSH) activity via BL detection of aminoluciferin conjugated to the chenodeoxycholic bile acid, prepared as BL probe. The intestinal microflora and microbiome-produced BSH play a central role in human health, but its function remains unclear due to the lack of suitable methods for measuring its activity.

## 2. Material and Methods

### 2.1. Chemicals

Luciferase from Photinus pyralis (EC 1.13.12.7) (specific activity $3 \times 10^7$ light unit/mg), D-luciferin, dithiothreitol (DTT), glutaraldehyde, proline, chenodeoxycholic acid, glycine, and aminoluciferin were purchased from Merck, Germany. Nylon-6 tubes were obtained from SNIA Viscosa, Milano, Italy. All other reagents and compounds were of analytical grade. All the solutions were made with apyrogenic reagent-grade water prepared with a Milli-Q System (Millipore Corp., Bedford, MA, USA). Cholylglycine hydrolase from Clostridium perfringens (C. welchii) lyophilized powder, ≥100 units/mg protein (one unit will hydrolyze 1.0 μmole of glyco-CDCA to glycine and CDCA in 5 min at pH 5.6 at 37 °C).

### 2.2. Apparatus

The scheme of the developed microfluidic devices is reported in Figure 1a, which illustrates the setup for ATP or BSH detection.

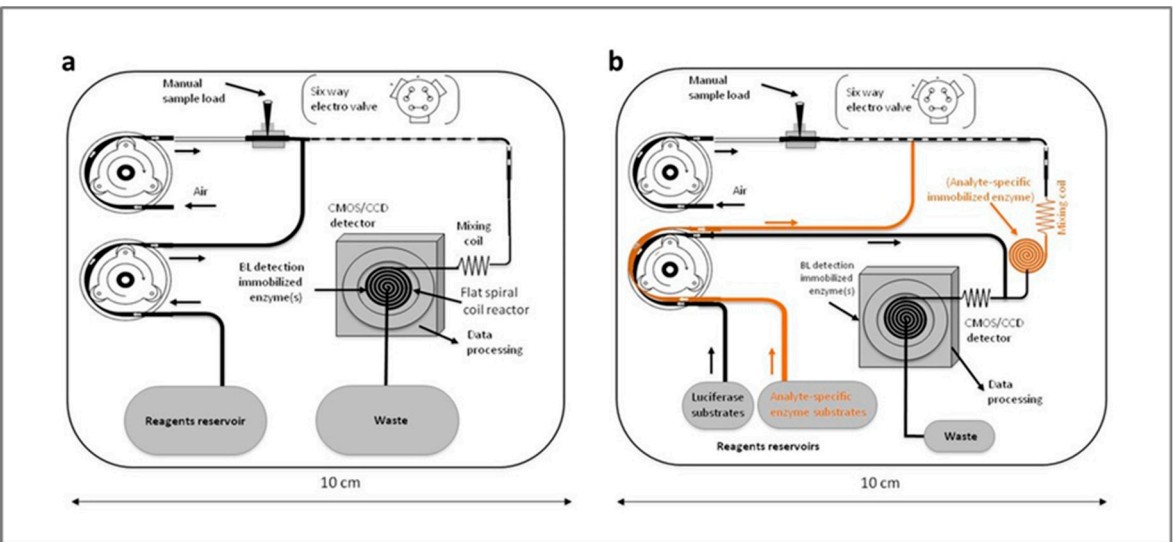

**Figure 1.** (**a**) Schematic representation of the continuous flow air segmented sensor for the detection of ATP or BSH enzymatic activity. (**b**) Generalized schematic representation of the device: additional reagents and analyte-specific enzymes can be added when coupled enzymes are used.

Two microperistaltic pumps (one/two channels each) drive the microfluidics to accurately deliver the solutions containing the substrates and reagents and the flow of air. The microfluidics is simple, and samples travel in 0.6 mm internal diameter tubes.

The sample solution (2–6 μL) is injected manually or via six-port compact electrovalve into the airflow stream. It travels between two bubbles of air and merges with the stream containing the reagents, resulting in a segmented flow of solution/air/solution. Immediately before detection, the flow stream passes through a debubbler mixing coil to remove the air bubbles. The flow stream reaches the flat coil spiral reactor with the immobilized BL/CL enzymes in the inner wall of the Nylon 6 tube with an 0.8 mm inner diameter and a length of 100–300 mm, as required by the enzyme catalytic activity and sample flow rate. The reactor is placed in contact with the light sensors. An additional reactor coil can be placed after the mixing coil, if one requires a further specific analyte enzyme that is not coimmobilized with the detection enzyme (due to partial compatibility in the pH and experimental conditions). In this case, a specific reagent stream must be used to deliver reagents to this analytical reactor (Figure 1b).

The passage of the sample through the spiral detector results in a BL light signal, which is imaged and recorded for a given time. The difference between the baseline and the maximum signal is related to the analyte concentration in the sample. The signal-recording mode is set up to collect all the emitted light over a fixed time, considering the baseline value.

By confining the stream between two air bubbles in the continuous air-segmented system, the setup prevents axial diffusion caused by the concentration gradient, which can reduce reproducibility. This ensures the precision and accuracy of the microfluidics with respect to conventional flow injection systems. This system also reduces carryover effects. A washing step can also be added after the detection to reduce the signal background.

### 2.2.1. Peristaltic Pumps

Two 25 mm diameter microperistaltic pumps (Figure 2) were used with 0.6 mm inner diameter silicon tubes. The flow rate was accurately controlled by a DC variable input. Different rates can be achieved by modifying the pump rotation speed and the tube's inner diameter. The second pump delivers air at a flow rate that is usually half that of the other stream. The sample solution is inserted by a simple T tube connector into this channel, which will merge into the solution stream containing all the analytical chemicals. The sample solution, travelling in a train of air bubble/sample solution/air bubble, is then

mixed in a spiral coil and reaches the flat spiral coil containing the immobilized specific CL enzymes placed in front of the CMOS/CCD detectors.

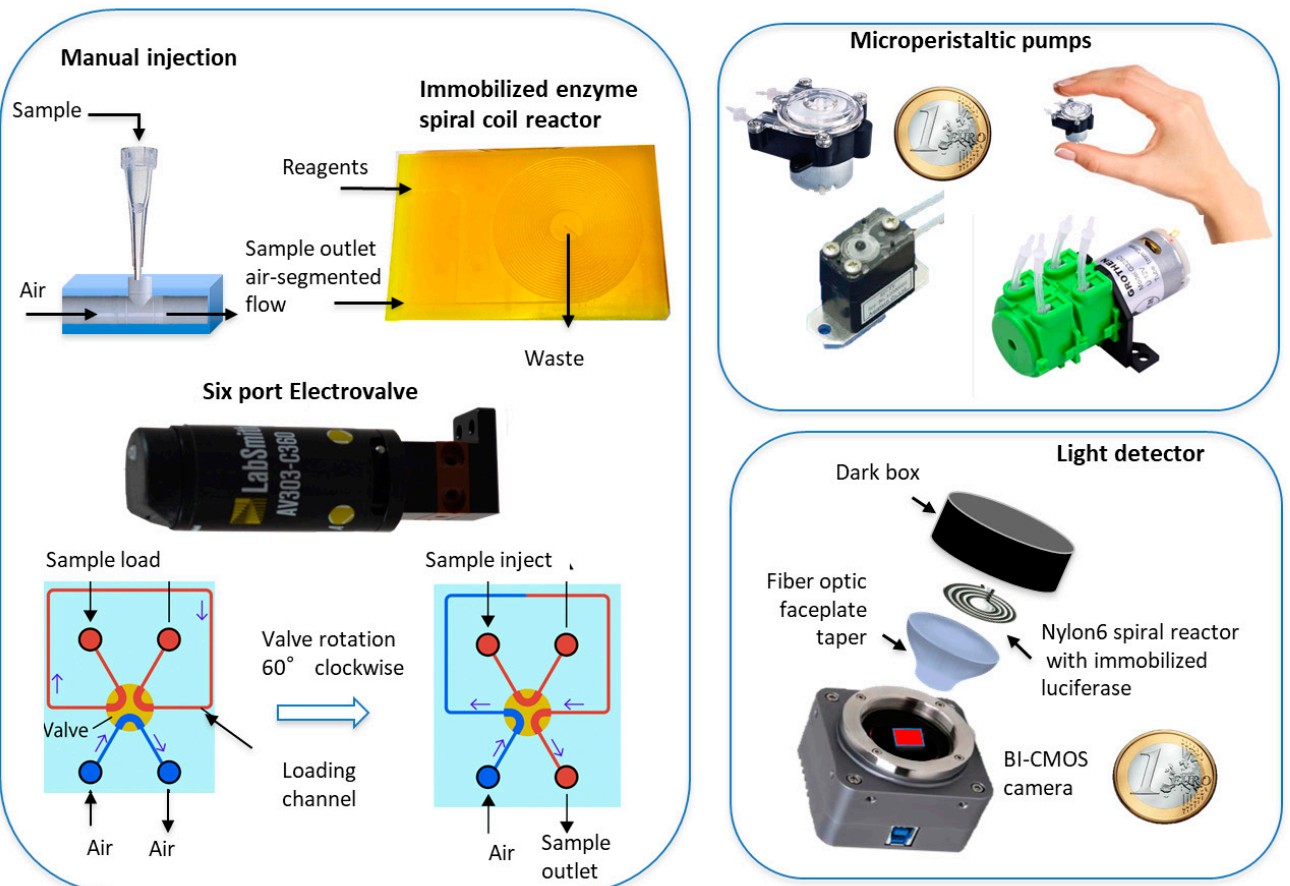

**Figure 2.** Continuous flow air segmented flow bioluminescence sensor main components.

When the second coil with analyte-specific immobilized enzyme is added, a peristaltic pump with two rotor heads generates two separate reagent streams. The stream with the substrates merges with the sample stream before the analyte-specific coil. The stream with the reagents for the detection system merges with the sample stream after the analyte-specific coil (Figure 1b).

### 2.2.2. Sample Injection and Reagent Delivery

The sample can be added manually or via a six-port injection valve (AV303, LabSmith, Inc., Livermore, CA, USA) for programmed automated sample injection (Figure 2). For the manual injection, we used a conventional pipette tip, usually 0.5–10 μL, connected tightly to a PDMS t-tube device, as previously described [6]. The tip acts as a mini-funnel, and the sample is aspirated by the airstream and transported to the substrate channel. The second peristaltic pump delivers the substrate and chemical solution for the biospecific reaction catalyzed by the immobilized luciferase. Its flow rate is usually higher than the airstream, which can be regulated by the voltage of the peristaltic pump or by the inner diameter of the silicon tube.

A rechargeable lithium battery can power the overall system including the CMOS/CCD, electrovalve, peristaltic pumps, and their relative speeds.

Sample can be intermittently added at a time interval of 2–6 min. A washing step can be included to eliminate carryover and reduce the background due to memory effect.

### 2.2.3. Firefly Luciferase Immobilization on Nylon 6

We selected Nylon 6 to immobilize enzyme because of our previous results showing stability and optimized activity [21].

Luciferase from Photinus pyralis was immobilized on Nylon-6 tube with 0.8 mm inner diameter. A flat spiral (25 mm diameter and 150 mm length) was formed by heating the tube at 80 °C for 15 min and rolling it to form the flat spiral coil (Figure 3). The immobilization procedure is a slight modification of our previously reported procedure [7].

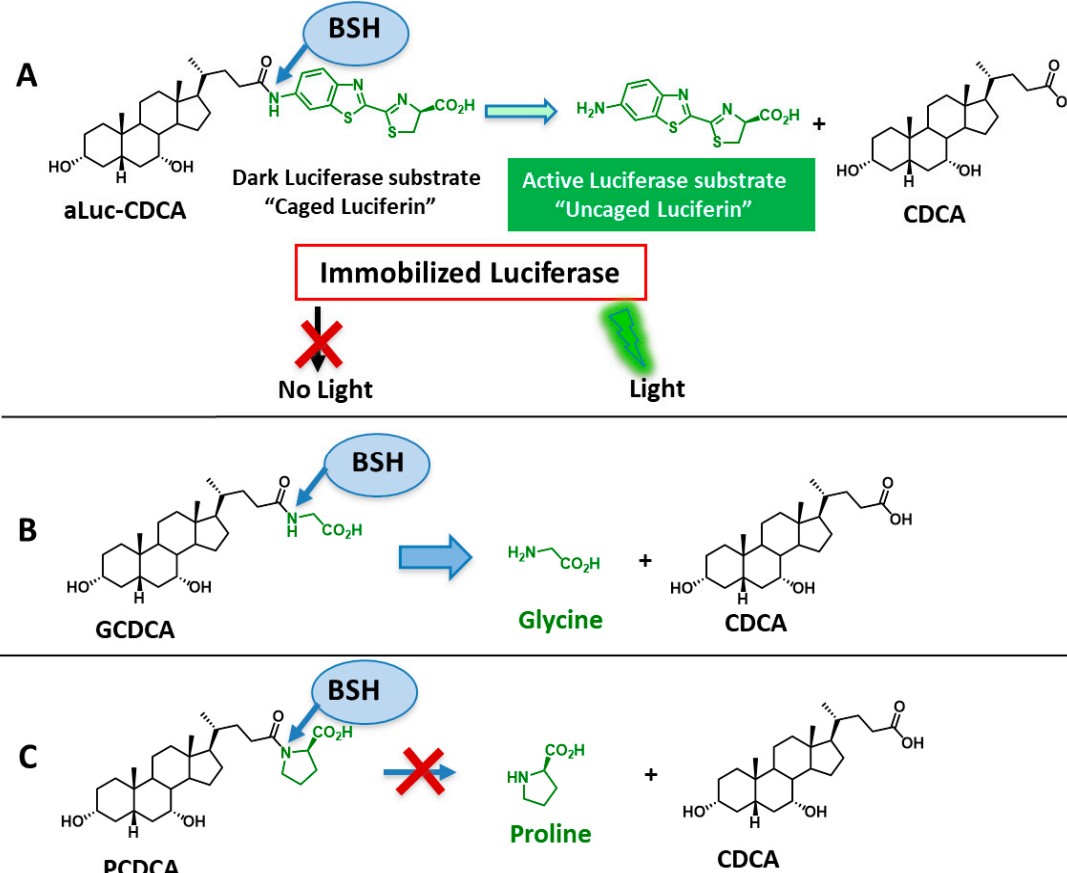

**Figure 3.** Schematic representation of the bioluminescence substrate mechanism for BSH activity measurement. (**A**) In the aLuc-CDCA probe, aminoluciferin is conjugated to chenodeoxycholic acid via an amide bond and therefore is not a substrate for luciferase. When released by the action of the BSH enzyme on the peptide bond, the free aminoluciferin becomes an active substrate for the luciferase and light will be emitted. (**B**) Metabolism of the natural occurring BA Glycochenodeoxycholic acid: GCDCA (**C**) Stable toward BSH hydrolysis, proline-conjugated CDCA (PCDCA).

Briefly, the Nylon 6 was O-alkylated by treatment with triethyloxoniumtetrafluoroborate for 5 min at 25 °C, washed with dichloromethane, filled with 1,6-diaminohexane in methanol (10%, $w/v$), and left to incubate for 45 min at 35 °C. After extensive washing with water, the tubes were activated by treatment with 5% ($w/v$) glutaraldehyde in 0.1 M borate buffer, pH 8.5, for 20 min at 25 °C. The tubes were washed with 0.1 M potassium phosphate buffer, pH 8, and filled with firefly luciferase (0.2 mg/mL) in 0.1 M potassium phosphate buffer, pH 8, 0.2 mM DTT, and 2 mM EDTA, and left to incubate overnight at 4 °C. The tubes were extensively washed with 0.1 M potassium phosphate buffer, pH 7, and stored in 0.1 M potassium phosphate buffer, pH 7, 1% bovine serum albumin, 1 mM DTT, and 0.02% sodium azide, at 4 °C. The amount of immobilized enzyme was calculated by subtracting the unbound enzyme activity from the total added activity.

The working bioluminescent solution for ATP quantification assay comprises 0.02 M Tris-acetate buffer, pH 7.8, containing 0.1 mM luciferin, 5 mM $Mg^{2+}$, 0.1 mM EDTA, and

1 °C in the dark. To measure BSH activity, we used the same solution without the luciferin but adding ATP 2 mM.

### 2.2.4. Light Imaging Detector

To image the BL light signal image, the microfluidic system includes an ultrasensitive B/W back-illuminated CCD, cooled with a double Peltier (if high detectability is required), or a BI-CMOS-based B/W or color photocamera. The flat spiral is placed directly in front of the detector in contact lensless mode using a fiber optic faceplate taper (25 mm, Edmund Optics, Barrington, NJ, USA), which enlarges the field of view focusing the spiral image and adapts the spiral's size to that of the detector (Figure 3). A glass mirror is placed behind the flat spiral to reflect all the emitted light toward the detector. The active measurement area is enlarged by a factor of 2.3 and the size of the CCD sensor increased from 9.0–6.7 mm$^2$ to 20.7–15.4 mm$^2$, permitting the use of a larger reaction spiral diameter with immobilized luciferase. The spiral and the detectors are enclosed in a cylindrical dark box to prevent any ambient light exposure.

The CCD imaging detector was built from a MZ-2PRO CCD camera (MagZero, Pordenone, Italy) equipped with a Sony ICX285 progressive scan monochrome CCD image sensor (1360 × 1024 pixels, pixel size 6.45 × 6.45 μm$^2$) and a 16-bit analog-to-digital (A/D) converter. To minimize the instrumental thermal noise, the CCD sensor is thermoelectrically cooled by a double Peltier cell. The cooling system is driven by an external control box, which can be powered by a 220 to 12 V ac/dc power adapter or a 12 V battery. The camera is controlled via a USB 2.0 interface by computer software used to set the camera operation parameters and for data acquisition. Images are recorded in the flexible image transport system (FITS) format. This digital file format is commonly used in astronomy and is compatible with most scientific image analysis software.

As an alternative to the CCD sensor (used for the ATP determination), a CMOS sensor can be used to further reduce the device's weight and size, although this slightly reduces the system's sensitivity and detectability. Many commercially available devices can be used according to the optimal quantum efficiency at the wavelength of the emitted light, which ranged from 520 to 600 nm when luciferin and aminoluciferin were used. For the BSH determination, we used a CMOS (Micron Imaging MT9M001, Micron Technology Inc., Boise, ID, USA) equipped with a 0.5-inch monochrome CMOS sensor (1280_1024 pixels) with a quantum efficiency of 52% at 550 nm. The device is much smaller than the cooled CCD and can be held between two fingers (Figure 3)

When D-luciferin was used (i.e., for ATP detection), light emission occurred at 520 nm. When aminoluciferin was used (i.e., to evaluate BSH activity), a slight shift to higher wavelength was observed at 580 nm. However, both wavelengths are optimal for the CCD and CMOS.

The BL signal was acquired for a total of 60 s. The image ws saved in TIFF format, then quantitative image analysis performed using the freeware software ImageJ v.1.46 (National Institutes of Health, Bethesda, MD, USA).

For each image, a region of interest (ROI) corresponding to the spiral coil was selected, and then the RGB values were computed and converted to HSV. The hue (H) value of the image acquired before sample injection (background signal) was then subtracted from the H value obtained for sample analysis.

### 2.3. *Microdialysis*

Microdialysis is a method to study in vivo the extracellular space based on the principle of diffusion. It can be used to measure various small molecules including ATP, as reported here.

Microdialysis derives from push-pull techniques where the perfusing solution circulates inside a semipermeable membrane instead of freely in tissue. It is a complementary approach to in vivo experiments in different organs.

The microdialysis probe comprises two concentric steel cannulas covered at the tip by a $2 \times 0.5$ mm dialyzing membrane (molecular mass cut-off around 20,000 Da). A physiological fluid is introduced through the inner cannula, flushes the inside of the membrane, and leaves via the outer cannula [22–24]. Rat rectal temperature was maintained between 36.5 and 37.5 °C. The probe was connected to a microinfusion pump and continuously perfused at 2 μL min-1 with Ringer solution. This system allows the sampling of endogenous substances from the extracellular space of many tissues and blood, and delivers exogenous substances to target areas. A microdialysis syringe pump was used to deliver the perfusate using PTFE tubing (0.25-mm inner diameter, 0.8-mm o.d.) [25]. Samples collected from the microdialysis were immediately injected into the microfluidic system. An electrovalve connected the microdyalis system directly to the flowstreams of the microfluidic device for on-line sample analysis, guaranteeing the continuous monitoring of parameters [26].

### 2.4. ATP Analysis

The developed manifold was first used for the bioluminescent continuous-flow determination of ATP from ex vivo samples, and for the in vivo study of rat brain samples collected via the microdialysis system. The flow system involved two streams: the first was the working bioluminescent solution (flow rate of 100 μL/min), the second was a continuous flow of air (flow rate of 50 μL/min) into which a known volume of sample perfusate (5 μL) was manually added intermittently through a micropipette tip, which was mounted vertically and acted as a funnel as previously described [7].

The microdialysis system was also used to collect samples to determine ATP in rat caudate nucleus, cerebrospinal fluid, and jugular vein as previously reported [27–30]. The dialyzed sample was collected, and the ATP content analyzed at 15-min intervals for 1 h by manual injections.

### 2.5. BSH Determination

BSH is an intestinal bacterial enzyme involved in host fat digestion and energy harvest [31]. Before determining BSH activity in biological fluids, we performed studies using CH, a member of the BSH family. This bacterial enzyme is able to fully hydrolyze the amide bond of all the physiological glycine-amidated and taurine-amidated BAs. It is therefore useful to compare the effect on the aminoluciferin-amidated and proline-amidated BA semisynthetic substrates developed as a BL probes.

#### 2.5.1. BSH BL Probes

An aminoluciferin-BA conjugated substrate for BSH was recently described. Its hydrolysis efficiency is comparable to that of naturally occurring glycine-conjugated and taurine-conjugated BAs [32].

Here, the aminoluciferin is released by a probe where it is caged by covalently binding to chenodeoxycholic acid (CDCA), which is a bile acid (BA). The aminoluciferin coupled to CDCA (aLuc-CDCA) is no longer a suitable substrate for the luciferase. When the action of BSH releases aminoluciferin from the amide bond, the uncaged aminoluciferin acts as a substrate for firefly luciferase. In the presence of ATP and Mg2+, light will be emitted (Figure 3).

The aLuc-CDCA compound is therefore used as a probe to evaluate the extent and efficiency of BA hydrolysis by bacteria microflora containing BSH-specific enzymes [33].

Moreover, we previously demonstrated that not all aminoacids bound to BAs behave like natural occurring glycine and taurine. For example, proline or compounds like N-methyl taurine or glycine (sarcosine) are very stable toward deamidation by the action of the BSH family. The lack of deconjugation is the result of the steric hindrance caused by N substituents, such as N-methyl groups, or the nitrogen involved in a cyclic ring for proline and its analogs [33]. Proline conjugated to CDCA (PCDCA) was used as negative control.

### 2.5.2. Synthesis of Bile Acid-Aminoluciferin-CDCA and Stable Amidated CDCA BSH Substrates

The BSH aminoluciferin-CDCA probe (aLuc-CDCA) was synthesized by reacting CDCA (1.2 equiv.) and ethyl chloroformate (3.0 equiv.) in the presence of triethylamine (3.5 equiv.) at 10 °C. The mixture was reacted overnight at room temperature, the resulting mixed anhydride was treated with aminoluciferin (1 equiv.) at 50 °C for 24 h. The adduct was purified by silica flash chromatography using DCM/MeOH (96:4, *v/v*) to give the desired aminoluciferin-CDCA derivative in 50% yield. Glyco-CDCA (GCDCA) and proline-CDCA (PCDCA) were prepared and fully characterized as previously described [33,34].

The data of the aLuc-CDCA were compared with the catalytic effect on physiological glycine conjugated CDCA (GCDCA). As a negative control which could not be conjugated and was stable toward cholyl glycine hydrolase (CH) (a member of BSH family deamidating enzymes), we used CDCA amidated with N-[(3α,5β,7α)-3,7, -dihydroxy-24-oxocholan-24-yl]-L-proline (PCDCA). PDA is in fact a cyclic amino acid, and the nitrogen atom in a cyclic ring prevented bacterial deamidation by steric hindrance to BSH [33].

### 2.5.3. BSH Enzymatic Activity

The activity of the BSH enzyme was evaluated with the aLuc-CDCA probe by the release of free semisynthetic aminoluciferin, which is not the ideal substrate for luciferase and ATP. Therefore, first we compared the results to conventional luciferin under the same experimental conditions by measuring ATP standards at different concentrations.

Then, the aLuc-CDCA probe and physiological GCDCA and the negative control PCDCA were incubated with the BSH to evaluate the kinetic profile of their hydrolysis [33].

The experiment was carried out in a solution consisting of acetate buffer (0.3 M, pH 5.6, 200 μL), EDTA (0.2 M, 40 μL), mercaptoethanol 0.2% (40 μL), enzyme (cholylglycine hydrolase, 50 μL, 100 units/mL), and GCDCA, PCDCA and aLuc-CDCA (10 mM, 10 μL). These were incubated at 37 °C and sample collected at 15-min intervals. The deconjugation of GCDCA and PCDCA was analyzed by HPLC-ES-MS/MS, as previously reported, using the procedure described below [35]. The release by BSH action of uncaged aminoluciferin was measured with the device using a BL cocktail similar to that used for ATP detection without the luciferin, which is the compound to be detected.

The metabolism of aLuc-CDCA evaluated by the amount of released free aminoluciferin with the BL method was compared with the HPLC-MS analysis of the CDCA produced.

### 2.5.4. Analysis of the Aminoluciferin Probe and Amidated BA Metabolism by HPLC-ES-MS/MS

The metabolism of glycine-conjugated and taurine-conjugated BAs in the presence of cholyl glycine hydrolase led to the formation of unconjugated BAs and the release of amino acid or aminoluciferin moiety (for the aminoluciferin-amidated BAs). In the presence of human intestinal bacteria in anaerobic conditions, the conjugated BAs are first deamidated to produce unconjugated BAs, then 7-dehydroxylated to be a BA substrate for the 7-dehydroxylase enzyme. The conjugated BAs (i.e., proline-conjugate, N methyl taurine) are stable in the intestinal microflora [36–38]. The metabolism of GCDCA, PCDCA, and aLuc-CDCA was evaluated by HPLC-MS as previously described [39].

### 2.5.5. BSH Ex Vivo Studies

Homogenized fresh human stools from healthy donors (500 mg) who gave consent to their use for research purposes, were transferred into sterile vials, to which 5 mL of sterilized chopped meat-glucose medium (Scott Lab., Fiskville, RI, USA) were added.

Samples were further diluted to obtain samples at different stool content from 0.1 to 2 g/L. BAs including GCDCA, PDCA, and aLuc-CDCA were then added at a final concentration of 50 mM and incubated at 37 °C. Then, at 15-min intervals for 2 h, the samples were analyzed to evaluate the extent of deamidation by measuring the free uncaged aminoluciferin released by the action of the enzyme present in human stools.

## 3. Results and Discussion

By exploiting recent advances in microfluidics technology, we have developed a compact device with high detectability and low cost for general use for any point-of-need sensitive continuous monitoring application. With respect to previously developed manifolds [6] based on air-segmented flow, this technology is simpler and suitable for out-lab use. We used miniaturized components such as miniperistaltic pumps, which have a rotor diameter of less than 25 mm and can accurately drive flow rate down to 100 μL/min. These offer many advantages for microfluidics, including no pump contamination by the fluids and easy cleanup. The low-cost microperistaltic pump also offers excellent repeatability in low-volume dispensing. The valveless design eliminates clogging and siphoning of liquid under most conditions. The relative flow rates of the two pumps were optimized to ensure an air-segmented flow, with the sample intermittently injected into the substrate solution and air. To reduce reagent waste, the inner diameter of the tubes can be smaller than in previous systems. However, the size should still allow bubble formation with a repeatable segmented flow. Inner diameters of less than 0.5 mm is therefore problematic due to capillary effects.

The sample volume for injection was properly optimized. In a tube with an inner diameter of 0.6 mm, a sample of 5 μL becomes a 17-mm cylinder in the stream, creating an air/water/air segmented flow where air bubble and sample have the same size. The optimal air flow rate is 50–80 μL/min, while that of the reagent is 100–120 μL/min in order to achieve the segmented flow. Under these conditions, the sample is immobilized for luciferase catalysis on the inner part of the Nylon 6 tube (diameter 0.8 mm) in the enzymatic reactor for 45 s. To guarantee the full catalysis of the substrate in the spiral, the rolled tube has a length of 150 mm in a flat spiral of 25 mm diameter, so it can be placed in contact with the imaging detector. The spiral length can be increased by preparing a two-layer flat spiral with a 300 mm tube length, which ensures the complete catalysis of the substrate in a time compatible with the overall flow rate and tube sizes. These experimental conditions are similar for ATP and BSH activity detection, except for the reagent cocktail. Under these experimental conditions, we can perform up to 10–20 analyses/hour, including a washing step.

The use of tubes with an inner diameter of 0.6–0.8 mm and a flow rate of 50–100 μL/min ensures reasonable reagent waste and optimal contact between the sample and the immobilized enzyme. The platform is ecosustainable thanks to the low reagent waste (less than 50 mL/h), the very diluted nontoxic solutions, and the use of reusable immobilized enzymes, with minimal plastics or other disposables.

### 3.1. Properties of Nylon 6 Immobilized Luciferase

The recovered activity of the Nylon 6 immobilized luciferase ranged from 0.8 to 1.5%, which is similar to previous results, while almost 90% of luciferase was immobilized [7]. The use of adipic acid dihydrazide as a spacer increased the enzymatic activity relative to that obtained with 1,6-diaminohexane, as a result of modifying the enzyme conformation induced by the positive charged group [40]. The Km value for the immobilized enzyme for ATP is $2.0 \times 10^{-5}$ M, which is similar to that of the enzyme in solution $1.8 \times 10^{-5}$ M. Despite the low activity of the immobilized luciferase, it was sufficient to fully catalyze the ATP or luciferin during contact. The main advantage of the immobilized enzyme is its relatively high stability, which is around 4 weeks at 25 °C and up to 10 weeks at 4 °C. For storage, the tubes should be filled with 0.1 M potassium phosphate buffer, pH 7, 1% serum albumin, 1 mM DTT, and 0.02% sodium azide as previously reported at 4 °C [7].

The Nylon 6 immobilized luciferase flat spiral was placed directly in contact in lensless mode with the CCD/CMOS sensors via tapered faceplate without any lens and focused by the mosaic of fiber optics (Figure 3). The clear image of the spiral ensures a proper collection of the emitted light. To enhance the light collection, the dark box cylindrical plug, which fully shields the detector from ambient light, has a mirror on the back surface and a white wall.

Imaging lasted for enough time to collect all the emitted light from the entire spiral. The acquisition time was fixed to 60 s. After this time, the emitted light output disappeared, indicating that all the substrate was catalysed by the enzyme and the system was ready for further measurements.

Moreover, the response of the semisynthetic aminoluciferin was slightly lower than that of luciferin. With an equimolar amount, the light emission is 40% lower.

As far as the detectors are concerned, despite a slightly lower detectability than a photomultiplier-based device [41], our system offers many portability advantages because the device can be powered by a lithium battery.

The continuous recording of the light emission imaging allows one to better define the optimized conditions to evaluate the time-dependent evolution of the light output and the completeness of the enzymatic reaction. The integral of the light emission over the beginning and end of the light signal allows one to define the analytical signal, which is compared with the blank signal before analysis or after the first run and before adding the next sample.

Relative to the cooled B/W CCD camera, the color CMOS is less sensitive by a factor of 100; however, it can image the color of the emitted light and so operate in spectra resolution mode to simultaneously measure probes with different emission wavelengths.

### 3.2. ATP Detection

The experimental parameters, flow rates, sample volume, and detector response were optimized to obtain the highest detectability in terms of precision and accuracy [7].

Figure 4 reports the dose-response curve for ATP detection. The dynamic range of linearity is from 0.01 to 100 nM concentration with a limit of quantification (LOQ) of 0.05 fmol injected ATP with a signal-to-noise ratio of 5:1. The sensitivity is 718 RLU nM$^{-1}$ allowing one to evaluate very small variations in ATP sample content.

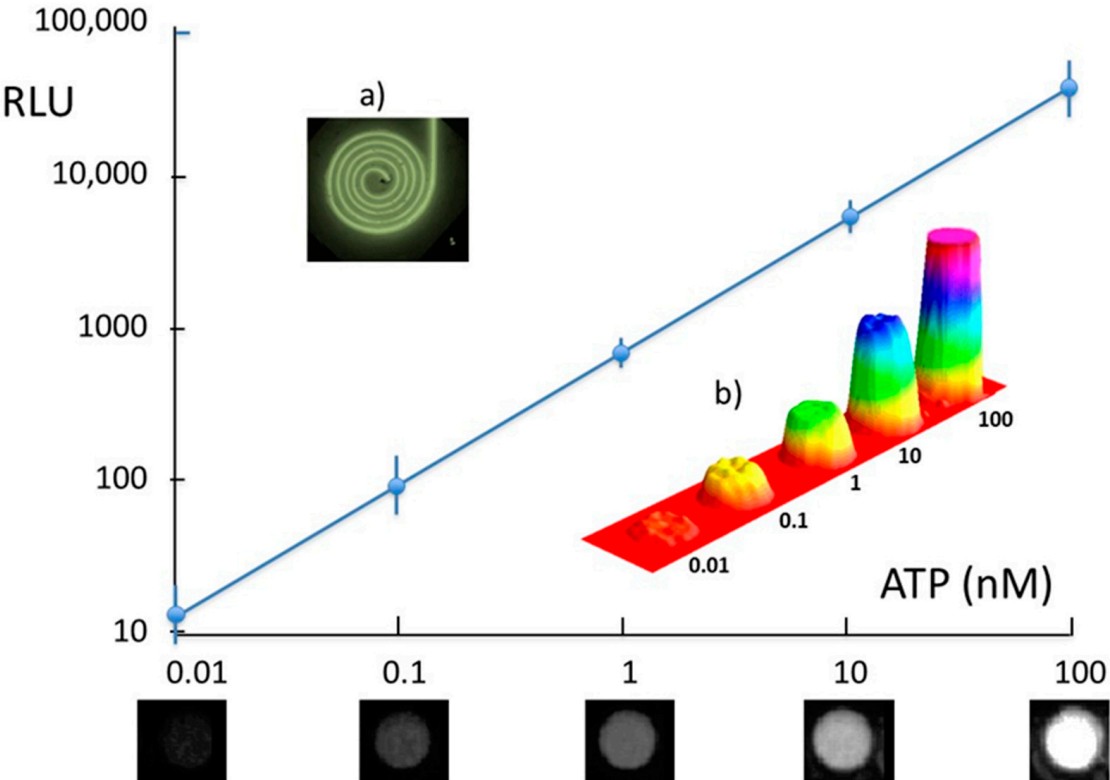

**Figure 4.** Dose-response curve for ATP. Mean value of 6 determination ±SD. (**a**) Example of light imaging from the flat spiral with immobilized luciferase using a color BI-CMOS. (**b**) 3D plot of the five sub-sequential analysis of the ATP standards. On the x-axis, the image of the light signal for the different ATP concentrations obtained with the cooled CCD detector.

The method has adequate precision and accuracy. The mean values and standard deviations of 10 replicates carried out at three ATP concentration levels of 0.1, 10, and 100 nM showed a CV% of less than 10%. The data were in agreement with a standard detection of ATP using commercial kits and microtiter reader, despite a slightly higher limit of detection with the CMOS, which can be reduced by two decades of concentration by using the cooled CCD. The air-segmented flow ensures high reproducibility with precision and a lack of carryover effect. An example of light imaging from the flat spiral with immobilized luciferase is also reported (Figure 4a). Figure 4b shows the 3D plot of the sequential ATP standard sample obtained using CMOS detector, where a background signal is achieved between each signal. The image of the light signal for the different ATP concentrations obtained with the cooled CCD detector are reported on x-axis, as an example.

Then, the biosensor was connected to microdialysis by injecting the perfusate manually. Table 1 shows the concentration of ATP in the perfusate of the studied organs and fluid.

**Table 1.** ATP concentration in perfusates collected from microdialysis.

| Body Fluid | ATP Conc Nmol/L | | | | |
|---|---|---|---|---|---|
| | **15** | **30** | **45** | **60** | **Min** |
| Jugular vein | 12 | 10 | 11.5 | 12 | |
| Cerebrospinal fluid | 800 | 745 | 873 | 701 | |
| Caudate nucleus | 9 | 11 | 9 | 12 | |

In the perfusate samples, the ATP concentration is very low in the caudate nucleus as expected, in the range of 0.45–0.6 fmol injected with minimal variation during the 1 h perfusion study. The ATP content is higher in the cerebrospinal fluid perfusate, and ten times higher in jugular vein. There were very low variations during the 1 h microdialysis study. These data are in agreement with previous work [42]. The developed biosensor could have useful applications, such as evaluating imbalance in extracellular ATP levels in brain tissue, which may be a triggering factor for several neurological disorders [43]. The developed microfluidic system setup is completely compatible with microdialysis systems and is suitable for a versatile hyphenation. A volume of 2–5 μL can be collected from microdialysis and its ATP content analyzed in a few minutes, allowing a near-continuous monitoring of ATP levels in extracellular fluid. A limit of the presented method for ATP measurement could be related to the delay from collecting sample to detection, which in some case may not reflect the real ATP dynamics. However, the microdialysis flows can be varied for clinical or preclinical studies to ensure that the concentration inside the perfusate is as close as possible to that in extracellular fluid, or if frequent sampling is needed. The frequency of sample analysis in the microfluidic system, the flow rate, and the signal acquisition are all adjustable.

### 3.3. BSH Activity

Treatment with BSH-positive probiotics or fecal transplants increases BSH activity in the gut. This increased BSH activity confers multiple health benefits including prevention against colon cancer and amelioration of the symptoms of Crohn's disease. [44,45]. The device was optimized for detectability and light signal reproducibility as a function of the reagent flow rate and immobilized luciferase activity.

First, the enzyme activity with the semisynthetic substrate was evaluated by directly analyzing increased amounts of amino-luciferin and conventional luciferin according to the expected concentration when released from caged amino-luciferin covalently bond to CDCA by the action of the BSH in real sample.

The dose-response curves obtained with amino-luciferin and luciferin are reported in Figure 5. These luciferins can be detected at as low as 0.5 μM (i.e 2.5 pmoles injected), with a lower limit of quantification (LOQ) for the luciferin. The aminoluciferin substrate produces a less sensitive dose-response curve with a slope of 4 RLU/μM versus 45 RLU/μM.

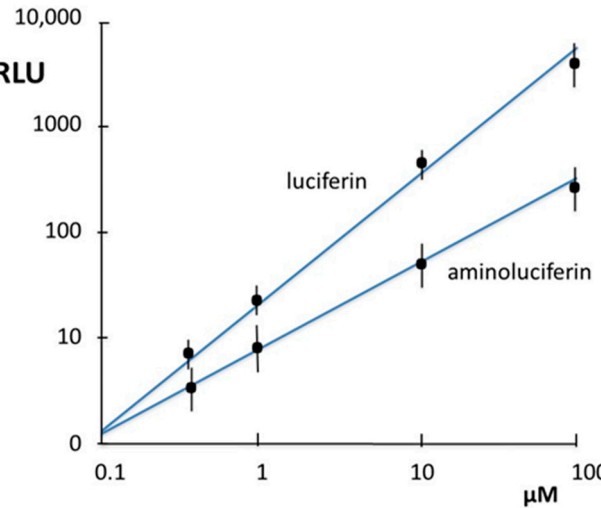

**Figure 5.** Calibration graph of luciferin and aminoluciferin. Mean values of 6 experiments ±SD.

In light of the analytical performance, we conducted an additional study using CH enzyme to hydrolyze both glycine-conjugated and taurine-conjugated BAs. The enzyme was used at a final activity in the tube of 15 U/mL, hydrolyzing the conjugates (aLuc-CDCA, GCDCA, PCDCA) in a few hours. In Figure 6, the metabolism of aLuc-CDCA probe measured in the microfluidic system is compared with that of GDCA, PCDCA, and aLuc-CDCA determined by HPLC-MS. When incubated with the enzyme, the GCDCA solution is quickly metabolized, and 50% of the probes are hydrolyzed after 26, 45, and > 120 min of incubation at 37 °C. The rate of biotransformation of the aLuc-CDCA is slightly lower but is still representative of the phenomenon. The negative control (PCDCA) is practically unmodified by the enzyme, with more than 98% recovered after 120 min.

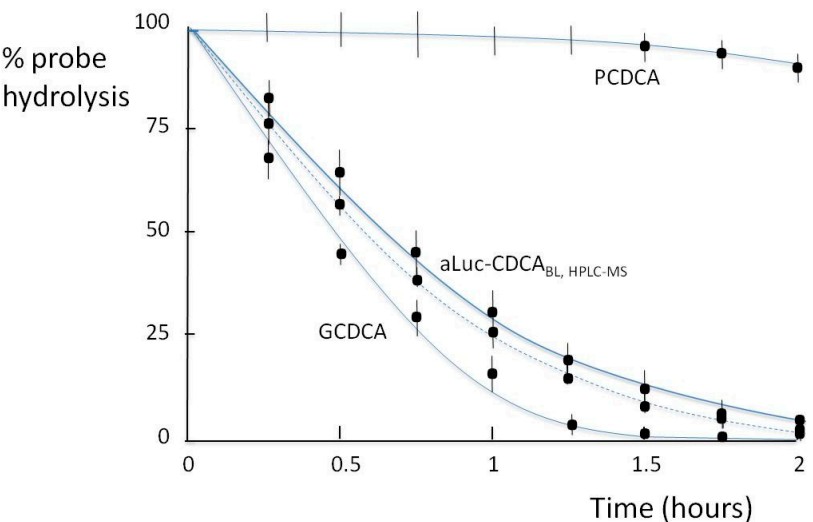

**Figure 6.** Metabolism of the aLuc-CDCA probe and GCDCA and PCDCA by the catalysis of BSH measured by HPLC/MS; metabolism of aLuc.CDCA probe measurement with the BL microfluidic system (dotted line).

Moreover, the results on metabolism of aLuc-CDCA evaluated by the amount of released free aminoluciferin with the BL method and the HPLC-MS analysis of the CDCA produced were in full agreement (Figure 6). Therefore aLuc-CDCA is suitable as a general probe to quickly evaluate the rate of BA deconjugation without more complex instrumentation such as HPLC-MS.

These data validate our probe for in vivo and ex vivo studies to evaluate variations in endogenous BSH activity in intestinal content as modulated by the microbiota and as induced by pathological conditions or by drug administration.

The aLuc-CDCA probe was used to evaluate BSH activity in human stools from a normal subject in anerobic conditions.

The rate of biotranformation of aLucCDCA and the other compounds was evaluated by incubation at 37 °C in the presence of different concentrations of the enzyme from 0 to 6 U/mL (Figure 7a) and with different amounts of fresh human stool homogenate concentration from 0 to 1.5 g/L (Figure 7b). The hydrolysis of GCDCA and PCDCA was evaluated by using HPLC-MS to measure the formation of free unconjugated CDCA, while that of aLuc-CDCA was evaluated with the BL microfluidic system in terms of the release of free aminoluciferin, as described in Material and Methods.

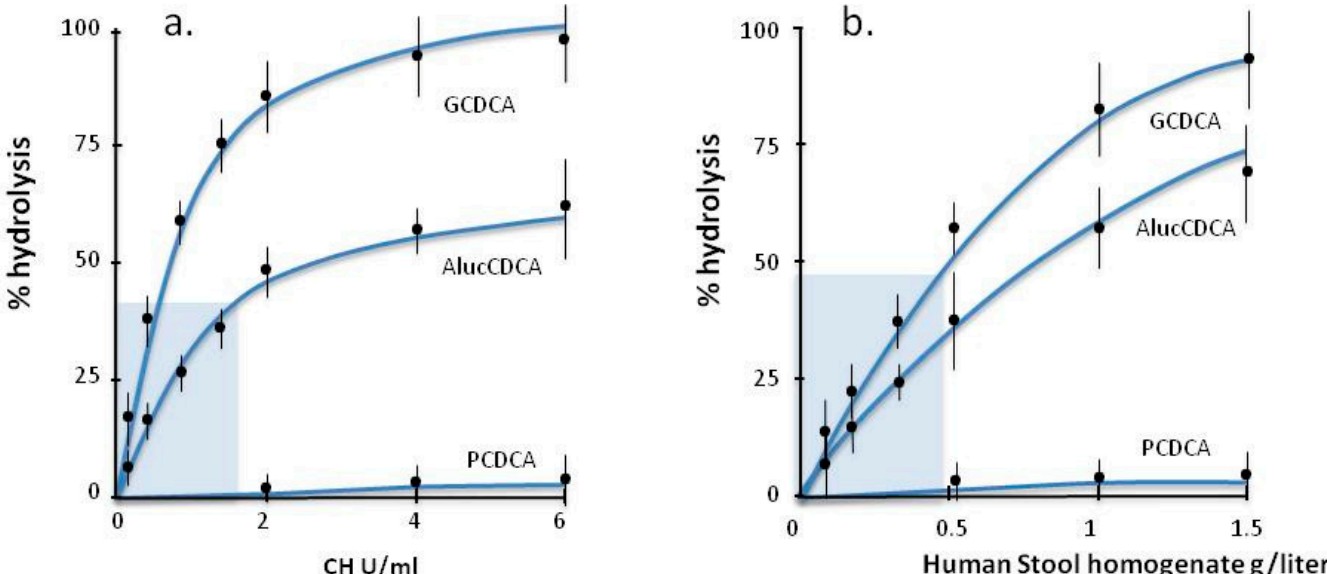

**Figure 7.** Mean values (six replicates ± SD) of kinetics hydrolysis of the different substrate biotransformation by the action of (**a**) Pure standard Enzyme Cholyglycine Hydrolase (CH); (**b**) Human stools homogenates at different dilutions. The light blue area represents the linearity range of % hydrolysis vs Enzymatic activity.

The data show a linearity of the light signal at a low enzyme activity (0.2–1.8 U/mL) and the kinetics is faster for the natural substrate GCDCA than for aLuc-CDCA, in line with previous data [33]. The incubation with different amounts of stool containing endogenous BSH showed a similar trend. These results demonstrate the efficacy in the use of aLuc-CDCA probe to measure BSH activity in real samples with a linearity range of 0.1–0.6 g/L for human stool homogenate. As reported in previous papers, the BL probes is highly specific toward BSH in biological samples. In fact, the amide bond of amidated BA is stable in duodenal fluid and bile and it can be hydrolyzed only by BSH enzyme [32,33]. The synthetic BL-probes should only present the requisite of a not hindered N position to allow the formation of amide bond, as it is for aLuc-CDCA, as described in Material and Methods.

The use of this method to define the complex microbiota along the entire intestinal tract could complement the recent in vivo imaging approach. Due to the analytical performances in term of linear and dynamic range, it could potentially be used in vivo during colonoscopy intervention [32,46] with minimal invasivity to study the microbiota and BSH activity along the entire intestinal tract; or it can be used for the quantification of BSH activity in clinical stool samples of patients [32].

## 4. Conclusions

The developed portable platform uses air-segmented flow and Nylon 6-tube-immobilized luciferase. It can simply and reliably assay picomolar amounts of ATP or aminoluciferin. Unlike packed columns, the nylon reactor had no problems with packing or disruption of the matrix or with bacteria growth, which markedly enhances the background light level.

The device can be made with low-cost components, including miniaturized microperistaltic pumps and CCD or CMOS light detectors, offering a simple analytical signal reading and processing. These components require minimal voltage and could potentially be powered by solar energy, making the system suitable for point of need use by non-specialist personnel.

The device has acceptable reproducibility, with CV % not greater than 15%, which can be reduced by using more expensive components. The device can process up to 20 samples per hour, using nylon tubes as a suitable support for luciferase. Despite the low activity of the immobilized enzyme, it was sufficient to achieve an extremely high sensitivity. With a single immobilized-enzyme coil, up to 900 ATP or aminoluciferin samples can be analyzed. Potentially, the method could also be used to assay a variety of other compounds or enzymes that are coupled to produce or consume ATP, including kinases like creatine kinase, which produces ATP from ADP [46].

The device can be extended to many biochemiluminescence-based segmented systems to generate a general microfluidic platform with easy replacement of the main luminescent reactor containing the immobilized indicator enzyme. For example, we can use immobilized peroxidase [13] to detect any compound that produces hydrogen peroxide when catalyzed by a specific oxidase enzyme such as glucose oxidase or lactose oxidase. The analyte-specific enzyme could be co-immobilized with peroxidase or in a separate coil placed on-line before the detector spiral. This will be a function of the enzyme's pH compatibility and the relative Km and catalytic activity.

Multichannel platforms can allow multiplex detection, thus offering potent analysis for efficient on-line sensitive detection.

Different luciferases can be used, including genetically modified firefly luciferase, nanoluc, and new luciferin-luciferase ultrasensitive systems. The same luciferase and different luciferins could lead to different wavelength light emissions, or different luciferin structures could be used. This allows multiplex systems, broadening the applicability.

The sample pretreatment is a limitation requiring further investigation. However, multicomponent analysis can be developed for on-site analysis.

The platform's components are cheap but offer adequate analytical performance. As a detector, the BI_CMOS of newer smartphone cameras can be used, as reported for other biosensor formats [46], allowing connectivity and low cost. The air-segmented priciple offers unique advantages in allowing sequential analyses without memory or carry-over effects. However, it cannot be further miniaturized, so other formats should be used for on-chip applications.

In conclusion, the continuous flow air-segmented device and flow injection platform in a user-friendly portable version allows continuous monitoring in ex vivo bioscience studies and in any process or screening control requiring portable devices.

**Author Contributions:** Conceptualization, A.R.; methodology, A.R., P.S., B.R., P.G., A.G.; validation, P.S., V.M., B.R.; formal analysis, V.M., P.G., G.M.; data curation, A.R., A.G., B.R.; writing—original draft preparation, A.R., P.G., P.S., V.M., G.M., A.G., B.R.; writing—review and editing, A.R., B.R.; supervision, A.R.; project administration, A.R.; funding acquisition, A.R. All authors have read and agreed to the published version of the manuscript.

**Funding:** This research received no external funding.

**Institutional Review Board Statement:** The study was approved by the Local Ethics Committee and followed the Declaration of Helsinki. Following written informed consent, stools were collected from volunteers who were recruited from the S. Orsola-Malpighi University Hospital, Bologna, Italy (study n 601/2018/Sper/AOUBo approved 20 February 2019).

**Informed Consent Statement:** Informed consent was obtained from all subjects involved in the study.

**Acknowledgments:** We thank Grace Fox for editing and proofreading the manuscript.

**Conflicts of Interest:** The authors declare no conflict of interest.

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
