# Peer review of "Compact Miniaturized Bioluminescence Sensor Based on Continuous Air-Segmented Flow for Real-Time Monitoring: Application to Bile Salt Hydrolase (BSH) Activity and ATP Detection in Biological Fluids"

_chemosensors, doi:10.3390/chemosensors9060122_

Round 1
Reviewer 1 Report
The authors described a flow-based bioluminescence detection method that is miniaturized to become a compact device while remaining an adequate performance for detection of the analytes, ATP and BSH activity using synthetic luciferin analog Aluc-CDCA. The result obtained from the prototype assay system seems promising because the miniaturization of the device retained adequate precision. However, there are several concerns regarding the description of the manuscript. Although they do not affect the main conclusion of the study, the points I made will be beneficial for avoiding potential misunderstanding of this research.
Major concerns;
- Comparison with the conventional method would be beneficial to characterize a newly developed assay system. The authors compared their newly developed bioluminescence assay with the HPLC-MS. However, they do not provide a direct comparison of the same analyte. Can the authors provide a result of AlucCDCA measured with HPLC-MS and compare the result of AlucCDCA measured with the developed assay system? Besides, for Figure7, the vertical axis represents RLU, however, this unit does not apply to GCDCA and PCDCA samples, because their quantities were measured by HPLC-MS. It is not clear how the authors converted the result of GCDCA and PCDCA to RLU. Thus, it might be misleading to compare with the results of ALucCDCA.
Minor concerns;
- When conducting an assay for the biological sample, selectivity is also an important factor for evaluating the assay system. Although the authors point out the importance of selectivity, they do not provide any evidence on the selectivity of the developed system. The authors should comment on the selectivity of the developed assay system.
- Some descriptions in the manuscript were placed at inappropriate section. In addition, some descriptions seemed to be repeated in the manuscript. Please consider the followings;
2-1. I would rather place the description of aLUC-CDCA (lines128-152) in the methods or results and discussion section because the description is too detailed.
2-2. “It uses many cheap…” (lines 165-168) are potential perspectives of the developed method, therefore I think it would be appropriate to be described in the result and discussion or conclusion section.
2-3. Lines 345-346 seem to be a repetition of the previous sentence.
2-4. Lines 379-382 and Lines 355-358 describe the same thing and seem to be redundant.
- For Figure4 a), it would be beneficial if the authors provide a bright-field image of the reactor spiral in order to support statements regarding lines 447-449.
- Some words had different written forms. Please consider them to be in a consistent form. For example, “i.d” and “inner diameters”, and “aLuc-CDCA”, “AlucCDCA”, and “ALucCDCA”.
- Some methodological descriptions lack information for a retest. For example, in Line256, what is the concentration of ATP?
- It is somewhat curious whether the authors provided appropriate references. For example, in Line85-Line107, the authors describes previous achievements of the field without any citations. Please cite references that the authors used to describe these descriptions.
Author Response
The Authors would like to thank Referee 1 for the revision. Figures 6 and 7 -related to the metabolism of the BL probe were adjusted and the text was modified accordingly; details were added to the manuscript based on reviewer comments.
Major concerns;
- Comparison with the conventional method would be beneficial to characterize a newly developed assay system. The authors compared their newly developed bioluminescence assay with the HPLC-MS. However, they do not provide a direct comparison of the same analyte. Can the authors provide a result of AlucCDCA measured with HPLC-MS and compare the result of AlucCDCA measured with the developed assay system? Besides, for Figure7, the vertical axis represents RLU, however, this unit does not apply to GCDCA and PCDCA samples, because their quantities were measured by HPLC-MS. It is not clear how the authors converted the result of GCDCA and PCDCA to RLU. Thus, it might be misleading to compare with the results of ALucCDCA.
Thank you very much for the criticism, it has been very useful to better clarify this point. The metabolism of the aLuc-CDCA probe was measured with the BL microfluidic system (through the BL detection of released amminocilucoferin) and also with HPLC-MS by the quantification of the CDCA produced. Of course, we can compare the results obtained for the aLuc-CDCA using the presented method to HPLC results, while the other two BA have been measured only with HPLC-MS. We better specified the approach in Material and Methods. We also modified Figure 6 including for aLuc-CDCA both the results obtained with the BL microfluidic system and HPLC-MS analysis.
We replaced the RLU y axis by % hydrolysis in Figure 7 and therefore modified the text accordingly.
Minor concerns;
- When conducting an assay for the biological sample, selectivity is also an important factor for evaluating the assay system. Although the authors point out the importance of selectivity, they do not provide any evidence on the selectivity of the developed system. The authors should comment on the selectivity of the developed assay system.
We agree and we further clarified this point by modifying the text accordingly. As previously reported [15] ans as we described in the text in par. 2.6.1., in BSH-BL probes the C24 position forms an amide bond with glycine and taurine. A similar behavior by BA can be observed with other aminoacids as long as the N is not hindered by an alkil group or a cyclic ring (as the one we introduced in the control reference compound). Of course, the aminoluciferin should behave like natural glycine/taurine even though strictly speaking the latter is not an aminoacid. The results in Figure 6 show a probe hydrolysis % similar to that of natural amidated-BA confirming the potential use of aLuc-CDCA as a BL-probe for BSH, with the only small difference being related to the Km toward the enzyme BSH.
Amidated-BAs are highly stable in duodenal fluid and bile and they were hydrolyzed only by BSH enzyme. This specificity was already discussed in previous papers. References and sentence on this point were added to the text (Discussion of results presented in Figure 7).
Some descriptions in the manuscript were placed at inappropriate section. In addition, some descriptions seemed to be repeated in the manuscript. Please consider the followings;
2-1. I would rather place the description of aLUC-CDCA (lines128-152) in the methods or results and discussion section because the description is too detailed.
We agree and we modified the text accordingly. The description of aLuc-CDCA was included in Material and Methods and paragraphs were reorganized.
2-2. “It uses many cheap…” (lines 165-168) are potential perspectives of the developed method, therefore I think it would be appropriate to be described in the result and discussion or conclusion section.
We agree and we modified the text. The comment was added to the Conclusion.
2-3. Lines 345-346 seem to be a repetition of the previous sentence.
Thanks. The sentence has been deleted.
2-4. Lines 379-382 and Lines 355-358 describe the same thing and seem to be redundant.
Thanks. In lines 355-358 we present the results reported in figure 5 for the comparison of conventional luciferin and aminoluciferin as luminescent substrates; while in line 379-382 we describe how to measure the hydrolysis of aLuc-CDCA and GDCA and PCDCA. We agree that we already anticipated this point in par “2.6 BSH determination”, so we eliminated the redundant sentence there and we modified the text in order to improve the description of the metabolism determination.
- For Figure4 a), it would be beneficial if the authors provide a bright-field image of the reactor spiral in order to support statements regarding lines 447-449.
The aim for light collection was the measurement of the light emitted from the entire spiral to enhance detection as much as possible and to capture all the signal from the enzymatic reaction; so 60 s was chosen as collection time. After this time, the light emitted disappears making the system ready for further analysis. The sentence in lines 447-449 was modified to stress this point on light detection.
- Some words had different written forms. Please consider them to be in a consistent form. For example, “i.d” and “inner diameters”, and “aLuc-CDCA”, “AlucCDCA”, and “ALucCDCA”.
Thank you for your remark. We modified in the text.
- Some methodological descriptions lack information for a retest. For example, in Line256, what is the concentration of ATP?
Thank you for your remark. The concentration value for ATP was added.
It is somewhat curious whether the authors provided appropriate references. For example, in Line85-Line107, the authors describes previous achievements of the field without any citations. Please cite references that the authors used to describe these descriptions.
References were added.
Reviewer 2 Report
Remarks to the authors:
The present study by Aldo Roda et al, entitled “Compact miniaturized bioluminescence sensor based on continuous air-segmented flow for real-time monitoring: application to bile salt hydrolase (BSH) activity and ATP detection in biological fluids” (###), demonstrated a concept of a portable device which could achieve out-lab use for detecting ATP concertation and BSH activity. The authors developed a compact device with high detectability and low cost for general use for any point-of-need sensitive continuous monitoring application.
ATP is an important and universal purinergic compound related in multiple biological processes by either intracellular energy supply or extracellular signaling. Also, BSH is an essential bile acids metabolism enzyme and reflects the human health status. By their crucial biological meanings, Dr. Roda and colleagues designed a portable, luciferase-based biosensor and showed its application in monitoring ATP concertation in different samples and BSH activity in human feces. In general, the manuscript is clearly written and the experiments are well conducted, however, a couple of modifications need to be made, and some issues need to be addressed and further discussed.
Major critiques
- In line 99-101, authors claimed the invasiveness of previous developed biosensors and the reference of previous methods are required. As the first application of this system in monitoring the ATP concentration in CSF, it is difficult to access to the CSF unless invasive approaches are performed to collect the fluid. And since the invasive approaches are required, the “out-lab use” of this system seems confusing. Authors could further discuss or make it less ambiguous.
- ATP is considered as a “danger signal” due to its ability in recruitment and activation of immune cells, and many extracellular enzymes are evolved to quickly convert ATP/ADP into adenosine. There are several minutes delay of this air-segmented system form collecting sample to detection, which may not reflect the real ATP dynamics.
- The ATP cc. detected in the jugular vein is unexpected high (around 10uM) considering the EC50 of P2Y12 and P2Y1 towards ATP/ADP are around hundred nmol range, activation of which are sufficient for platelet aggregation. Previous study who stabilized ATP in plasma by preventing its degradation or leakage from cells dropped into the conclusion that extracellular ATP cc. is quite low in the blood (DOI: 10.1373/clinchem.2006.076364), different from results in this paper. These are opposite results that authors may want to consider or discuss.
- The specificity of BSH activity assay needs further discussion and additional experiments could be conducted. Since there other enzymes with similar functions, such as PA, may also hydrolyze luciferin substrate
Minor critiques
- Line 558-562, the linearity range of this aLuc-CDCA probe is 0.1-0.6 g/L when measuring BSH activity in human samples. First of all, the health status of the volunteers who donated feces sample is better to be clarified. Secondly, whether this range is suitable for general application cause some patients may got higher BSH level?
- Line 27 The end of the sentence has a redundant full stop.
- The Fig2 consists of 2 panels with similar contents, which seem to be redundant. We suggest the authors integrate them into one panel.
Author Response
Reviewer 2
Authors want to thank Referee 2 for the revision. All the comments were accepted and the revisions made according.
The present study by Aldo Roda et al, entitled “Compact miniaturized bioluminescence sensor based on continuous air-segmented flow for real-time monitoring: application to bile salt hydrolase (BSH) activity and ATP detection in biological fluids” (###), demonstrated a concept of a portable device which could achieve out-lab use for detecting ATP concertation and BSH activity. The authors developed a compact device with high detectability and low cost for general use for any point-of-need sensitive continuous monitoring application.
ATP is an important and universal purinergic compound related in multiple biological processes by either intracellular energy supply or extracellular signaling. Also, BSH is an essential bile acids metabolism enzyme and reflects the human health status. By their crucial biological meanings, Dr. Roda and colleagues designed a portable, luciferase-based biosensor and showed its application in monitoring ATP concertation in different samples and BSH activity in human feces. In general, the manuscript is clearly written and the experiments are well conducted, however, a couple of modifications need to be made, and some issues need to be addressed and further discussed.
Major critiques
- In line 99-101, authors claimed the invasiveness of previous developed biosensors and the reference of previous methods are required. As the first application of this system in monitoring the ATP concentration in CSF, it is difficult to access to the CSF unless invasive approaches are performed to collect the fluid. And since the invasive approaches are required, the “out-lab use” of this system seems confusing. Authors could further discuss or make it less ambiguous.
References on sensor and systems for in-vivo or ex-vivo measurement were added to the Introduction. We discussed in the Introduction the use of miniaturized systems for the monitoring of in-vivo parameters. These systems may allow for a simpler and more direct implantation to improve sample recovery and as a consequence sensitivity of the analysis. Moreover, these systems are suitable for in-situ analysis, i.e. animal facilities, with improvement of analytical performances. We added comments on this point to the Introduction.
- ATP is considered as a “danger signal” due to its ability in recruitment and activation of immune cells, and many extracellular enzymes are evolved to quickly convert ATP/ADP into adenosine. There are several minutes delay of this air-segmented system form collecting sample to detection, which may not reflect the real ATP dynamics.
We agree with the referee and we included this issue and limitation in the text to better clarify this point in the discussion on results for ATP quantification.
- The ATP cc. detected in the jugular vein is unexpected high (around 10uM) considering the EC50 of P2Y12 and P2Y1 towards ATP/ADP are around hundred nmol range, activation of which are sufficient for platelet aggregation. Previous study who stabilized ATP in plasma by preventing its degradation or leakage from cells dropped into the conclusion that extracellular ATP cc. is quite low in the blood (DOI: 10.1373/clinchem.2006.076364), different from results in this paper. These are opposite results that authors may want to consider or discuss.
We thank the reviewer for the comment. There was a mistake in the data reported in Table 1 for jugular vein. The ATP concentration is around the dozens nmol. We corrected the values.
- The specificity of BSH activity assay needs further discussion and additional experiments could be conducted. Since there other enzymes with similar functions, such as PA, may also hydrolyze luciferin substrate
We agree and we further clarified this point by modifying the text accordingly. As previously reported [15] and as we described in the text in par. 2.6.1., in BSH-BL probes the C24 position forms an amide bond with glycine and taurine. A similar behavior by BA can be observed with other aminoacids as long as the N is not hindered by an alkil group or a cyclic ring (as the one we introduced in the control reference compound). Of course, the aminoluciferin should behave like natural glycine/taurine even though strictly speaking the latter is not an aminoacid. The results in Figure 6 show a probe hydrolysis % similar to that of natural amidated-BA confirming the potential use of aLuc-CDCA as a BL-probe for BSH, with the only small difference being related to the Km toward the enzyme BSH.
Amidated-BAs are highly stable in duodenal fluid and bile and they were hydrolyzed only by BSH enzyme. This specificity was already discussed in previous papers, references and sentence on this point were added to the text (Discussion of results presented in Figure 7).
Minor critiques
- Line 558-562, the linearity range of this aLuc-CDCA probe is 0.1-0.6 g/L when measuring BSH activity in human samples. First of all, the health status of the volunteers who donated feces sample is better to be clarified. Secondly, whether this range is suitable for general application cause some patients may got higher BSH level?
Thank you for your remark, we added details to the text about stool samples.
BSH plays a central role in human health. Cost-effective quantification of BSH activity shows interesting perspectives for several biological assays to study its activity in the gastrointestinal tract. Examples are the study of the increase of BSH activity levels after the treatment with probiotics or fecal transplant with reduction in inflammation, amelioration of the symptoms and improvement of digestive functions. The presented BL-microfluidic system shows analytical performances (linearity and dynamic range) suitable for these applications. For example, given the published data [15], the microfluidic system could be used for the quantification of BSH activity in clinical stool samples of patients with inflammatory bowel disease.
Comments were added on the text (comments on results presented in Figure 7)
- Line 27 The end of the sentence has a redundant full stop.
Thank you, we deleted it.
- The Fig2 consists of 2 panels with similar contents, which seem to be redundant. We suggest the authors integrate them into one panel.
In Figure 2a the schematic representation of the continuous flow air segmented sensor for the detection of ATP or BSH enzymatic activity is shown with the sample injection and reagents for the BL reaction; while, in figure 2b a generalized schematic representation of the device with: additional reagents and analyte-specific enzymes when coupled enzyme reaction are used for analyte detection. We tried to integrate into one panel, however the schemes were not clear. We decided to arrange the images and compact them into a single frame (in the revised version of the paper as Figure 2) with two panels, to better highlight the differences on the two configurations.
Round 2
Reviewer 2 Report
Authors have fully addressed our questions in the revised manuscript and we believed this paper is qualified to be published on Chemosensors.